# Progressive, Qualitative, and Quantitative Alterations in HDL Lipidome from Healthy Subjects to Patients with Prediabetes and Type 2 Diabetes

**DOI:** 10.3390/metabo12080683

**Published:** 2022-07-25

**Authors:** Christina E. Kostara, Kiriaki S. Karakitsou, Matilda Florentin, Eleni T. Bairaktari, Vasilis Tsimihodimos

**Affiliations:** 1Laboratory of Clinical Chemistry, Faculty of Medicine, University of Ioannina, 45500 Ioannina, Greece; chkostara@gmail.com (C.E.K.); kikosts@yahoo.gr (K.S.K.); ebairakt@uoi.gr (E.T.B.); 2Department of Internal Medicine, Faculty of Medicine, University of Ioannina, 45110 Ioannina, Greece; matildaflorentin@yahoo.com

**Keywords:** HDL, lipidomics, NMR spectroscopy, prediabetes, type 2 diabetes

## Abstract

Prediabetes is a clinically silent, insulin-resistant state with increased risk for the development of type 2 diabetes (T2D) and cardiovascular disease (CVD). Since glucose homeostasis and lipid metabolism are highly intersected and interrelated, an in-depth characterization of qualitative and quantitative abnormalities in lipoproteins could unravel the metabolic pathways underlying the progression of prediabetes to T2D and also the proneness of these patients to developing premature atherosclerosis. We investigated the HDL lipidome in 40 patients with prediabetes and compared it to that of 40 normoglycemic individuals and 40 patients with established T2D using Nuclear Magnetic Resonance (NMR) spectroscopy. Patients with prediabetes presented significant qualitative and quantitative alterations, potentially atherogenic, in HDL lipidome compared to normoglycemic characterized by higher percentages of free cholesterol and triglycerides, whereas phospholipids were lower. Glycerophospholipids and ether glycerolipids were significantly lower in prediabetic compared to normoglycemic individuals, whereas sphingolipids were significantly higher. In prediabetes, lipids were esterified with saturated rather than unsaturated fatty acids. These changes are qualitatively similar, but quantitatively milder, than those found in patients with T2D. We conclude that the detailed characterization of the HDL lipid profile bears a potential to identify patients with subtle (but still proatherogenic) abnormalities who are at high risk for development of T2D and CVD.

## 1. Introduction

The prevalence of prediabetes, the intermediate metabolic state between normal glucose homeostasis and type 2 diabetes (T2D), [1] is increasing rapidly worldwide. This clinically silent, insulin-resistant state that often remains undiagnosed is associated with an increased risk for the development of T2D [2] and cardiovascular disease (CVD) [3]. In addition, observational evidence shows an association of prediabetes with complications such as nephropathy, neuropathy, and retinopathy that are traditionally considered consequences of established T2D [4,5]. Although multiple factors have been involved in the pathogenesis of prediabetes [6], the pathophysiologic mechanisms underlying the progression of prediabetes to overt T2D and also the proneness of these patients to developing premature atherosclerosis remain obscure.

Disturbances in glucose homeostasis and lipid metabolism, both well recognized predisposing factors for cardiovascular disease, are highly intersected and interrelated at multiple metabolic crossroads in different tissues [7]. Thus, abnormalities in pathways involved in lipoprotein metabolism could at least partially explain the risk for the development of atherosclerosis at this early stage of carbohydrate-metabolism derangement [3].

An atherogenic serum lipid profile hallmarked by high levels of triglycerides and low levels of high-density lipoprotein cholesterol (HDL-C) is often prevalent in patients with impaired glucose metabolism [8]. The cholesterol content of HDLs is used in clinical practice for the assessment of their atheroprotective capacity. Nevertheless, low or high HDL-C levels alone fail to accurately reflect the functional status of these particles. HDL functionality is inextricably related to their overall complex structure and composition, therefore qualitative and quantitative modifications in lipid compositional characteristics may lead to the conversion of atheroprotective HDLs into proatherogenic equivalent with impaired biological functions [9,10]. Apart from the well established protective function of HDLs against atherosclerosis, these particles directly affect glucose metabolism by inhibiting the stress-induced death of the pancreatic β-cells and enhancing glucose-stimulated insulin secretion [11,12]. However, the molecular components mediating these protective effects of HDLs against the apoptosis of pancreatic beta cells have not yet been investigated.

An in-depth characterization of HDL compositional characteristics with omics approaches such as lipidomics in individuals with prediabetes may contribute to the identification of early atherosclerotic signs not captured by the standard lipid parameters determined in everyday routine. Technological advances in analytical methods such as mass spectrometry (MS) and nuclear magnetic resonance (NMR) spectroscopy have facilitated efforts to unravel the metabolic dysregulation in complex lipid-related disorders and to identify predictive biomarkers beyond traditional lipids [13].

The presence of dysfunctional and compositionally altered HDL particles has been described in T2D [14,15,16]; however, to the best of our knowledge, alterations in HDL lipidome in individuals with prediabetes have not yet been investigated. In the present study, we investigated the HDL lipidome in patients with prediabetes and compared it to that of normoglycemic individuals and patients with established T2D.

## 2. Results

***Characteristics of the study’s participants***: The main demographic and biochemical data of the three groups studied are summarized in Table 1. As described in the Subjects and Methods section, all groups studied were well matched for age, gender, and serum-lipid-profile parameters to minimize their confounding effect on the analysis. As anticipated, fasting glucose levels and HbA1c were found in higher levels in patients with prediabetes and T2D compared to the normoglycemic control group (*p* < 0.001 in both diseased groups).

***HDL Lipidome:***Table 2 displays the percentage of the major surface and core lipid molecules of HDLs (cholesterol (total, free, and esterified), phospholipids, triglycerides), and the total core and surface lipids, as well as their ratios in the three groups studied and their % change in the pairwise comparison.

***Major lipid classes***: Patients in the prediabetes group displayed a significantly higher percentage of HDL core lipids as expressed by the sum of cholesterol ester (CE) and triglycerides (TG), mainly due to the significantly higher percentage of TG (increased by approximately 27%) compared to normoglycemic individuals (Table 2). In prediabetes, HDLs appear enriched in cholesterol, mainly due to a significantly higher percentage of free cholesterol (FC) (an increase of approximately 16%), while CE that is formed from FC by the action of the lecithin: cholesterol acyltransferase (LCAT) enzyme was lower but without significance. These qualitative aberrations in the fundamental structural components of HDLs found in the prediabetes vs. normoglycemic comparison were further deteriorated towards the same direction in T2D patients. Thus, TC, FC, and TG were further increased, whereas CE were further decreased (Table 2).

The aforementioned changes resulted subsequently in a progressive decrease in CE/TG (Table 2) and CE/FC ratio (4.26 ± 0.42, 3.76 ± 0.70, 2.45 ± 0.44, *p* < 0.001 for all comparisons, Figure 1b) in normoglycemic to prediabetes and then to T2D. The later was calculated from the well resolved signal of carbon-18 appearing at 0.68 ppm in the proton NMR spectrum, and it is further split into its two forms, free and esterified (Figure 1a) [14]. It is worth noting that the ratio CE/total cholesterol (CE/TC) in HDLs is significantly decreased with the progressive impairment of glucose metabolism (Figure 1c) (0.81 ± 0.02, 0.79 ± 0.03, 0.71 ± 0.04, *p* < 0.001 for all comparisons).

Total HDL surface lipids expressed as the sum of FC and total phospholipids (PLs) was lower in prediabetes due to the significantly lower percentage of total PLs (GPLs, ether GLs, SLs), which was further decreased in T2D resulting in a significant and gradual increase of TC/PLs ratio (Table 2).

***Phospholipid profiling***: Table 3 shows the changes in the percentages of the total amount of glycerophospholipids (GPLs), ether glycerolipids (ether GLs), and sphingolipids (SLs), as well as individual phospholipid molecules, quantified from characteristic well resolved signals in the proton NMR fingerprint, and their % change in the pairwise comparison.

The significantly lower percentages of total GPLs and total ether GLs are the main cause of the lower percentage of total PLs observed in prediabetes compared to normoglycemic subjects, while total SLs were significantly higher (Table 3). It is notable that these changes were further deteriorated towards the same direction in T2D patients.

The profile of individual PLs in prediabetes was different from that in the normoglycemic group, as shown in Table 3. Phosphatidylcholine (PC), the predominant phospholipid in HDL, presented a decrease in patients with prediabetes compared to controls. The percentages of phosphatidylethanolamine (PE), phosphatidylinositol (PI), and the rest GPLs, mainly attributed to phosphatidylserine (PS) and cardiolipin, were significantly lower in prediabetes compared to normoglycemic subjects, while notably lysophosphatidylcholine (LysoPC) was increased significantly, approximately by 62%.

For the ether GLs, the percentage of their total amount was lower in prediabetes compared to the normoglycemic group due to the lower percentage of those possessing an ether linkage (alkyl lipids) at the sn1 position of the glycerol backbone, the rest ether GLs mainly attributed to the platelet-activating factor (PAF), while those possessing a vinyl ether linkage at the sn1 position of the glycerol backbone or plasmalogens were not significantly different between the two groups (Table 3).

Finally, as seen in Table 3, the percentage of the total SLs was higher in prediabetes compared to controls mainly due to a significant increase of the rest SLs attributed to ceramide, sphingosine-1-phosphate (S1P). The percentage of sphingomyelin (SM), the second-most abundant phospholipid on the particle’s surface was significantly lower in prediabetes. The above-described changes in the main HDL-PLs resulted in alterations of their proportion, as expressed by the molecular ratio PC/SM. As seen in Table 3, PC, SM, rest GPLs, and rest ether GLs were further decreased in T2D compared to prediabetes, whereas rest SLs were further increased.

***Fatty acid pattern***: The fatty acid compositional pattern of esterified lipids carried by HDLs is shown in Table 4. Patients with prediabetes presented with a significantly higher percentage of saturated fatty acids (SFA) (an increase of approximately 13%) and a lower percentage of unsaturated ones (a decrease of approximately 8%) compared to the control group. It is worth mentioning that this decrease is mainly due to monounsaturated fatty acids (MUFA) with a significance of <0.001, whereas polyunsaturated fatty acid content is lower but without significance (Table 4). In individual fatty acids, this trend was statistically significant only in docosahexaenoic acid and the sum of arachidonic and eicosapentaenoic acid. In a T2D vs. prediabetes comparison, the fatty acid pattern shifted towards a more atherogenic profile with the further replacement of unsaturated fatty acids (UFA) (monounsaturated and polyunsaturated) with saturated ones (Table 4). Individual polyunsaturated fatty acids (PUFA) were also decreased with the exception of docosahexaenoic acid, which was found to be higher in T2D (Table 4). The above-described changes in the fatty acid pattern resulted in significantly higher ratios of saturated to unsaturated fatty acids and saturated to polyunsaturated fatty acids with the progressive impairment of glucose metabolism (Table 4).

The aforementioned progressive qualitative and quantitative alterations in the HDLs’ lipid compositional characteristics in normoglycemic to prediabetes and then to T2D patients are depicted in Figure 2a and better in the OPLS-DA scores plot (Figure 2b) created with untargeted analysis, suggesting a relatively high impact from the impairment of glucose metabolism on HDL lipid composition. The separation among three groups assessed by the following quality parameters of the resulting OPLS-DA model: R2X = 0.607, R2Y = 0.593, and Q2Y = 0.519, and the CV-ANOVA *p*-value was <0.001. Finally, additional pair-wise OPLS-DA models (normoglycemic group vs. prediabetes, prediabetes vs. diabetes, and normoglycemic group vs. diabetes) were constructed (Appendix A).

## 3. Discussion

In the present study, we investigated the HDL lipidome of patients with different degrees of glucose tolerance using NMR spectroscopy. Even though HDL-cholesterol levels were comparable in the groups studied, patients with prediabetes presented significant compositional alterations in HDLs’ lipid cargo compared to normoglycemic individuals and T2D patients. Of note, the changes that occurred in prediabetes were qualitatively similar but quantitatively milder than those that occurred in T2D. To the best of our knowledge, there are few data in the literature on the lipoprotein profile of individuals with different degrees of impaired glucose metabolism (i.e., prediabetes or T2D), which mainly concerns the investigation of plasma lipid profile [17] or lipoprotein subclass profile [18,19].

Many of the changes in HDL metabolism in insulin-resistance states have been pathophysiologically linked to elevated serum TG levels [20,21]. In our study, a significant and gradual TG enrichment in HDLs’ core from normoglycemic, in prediabetes, and then in T2D patients was observed, notwithstanding the fact that serum TG levels were within normal range and comparable among the three groups. This abnormality, together with the simultaneous depletion in CE, resulted in a gradual decrease of the CE/TG ratio, which is considered a critical factor for apoAI conformation, particles’ stability, plasma residence time, and also HDLs’ functionality [22,23]. In addition, a low CE/TG ratio is a feature of glycated LpA-I HDLs [24], suggesting that alterations in core composition may parallel HDL apolipoprotein glycation.

The enhancement of CE transfer rates mediated by CE transfer protein (CETP) is associated with TG enrichment and CE depletion as seen in our patients [25,26,27]. The resulting HDLs are intrinsically more unstable [22,28] and are undergoing rapid catabolism through the hydrolysis of TG by hepatic lipase [28]. Hepatic lipase activity is increased in insulin-resistance states, although the underlying mechanisms are not yet understood [29]. Notably, these HDLs may have attenuated or impaired antioxidative activity because the replacement of CE with TG considerably alters the quantity and quality of apoAI and especially the conformation of the central and C-terminal domains that are critical for HDL to act as an acceptor of LDL-derived oxidized lipids [22,23]. These particles exhibit increased susceptibility to oxidation [30] and the HDL-induced endothelial nitric-oxide synthase (eNOS) activation is decreased [31]. The aforementioned abnormalities constitute a key mechanism involved in the deficient HDL-cholesterol efflux capacity in T2D. In is worth noting that, in our study, these alterations were apparent early, in the prediabetes state.

HDLs may enhance glucose uptake in skeletal muscle [32,33,34]. Experimental studies have shown that apoAI promotes glucose uptake in skeletal muscle by a mechanism similar to that of the insulin, which involves translocation of GLUT4 glucose transporters [35]. This hypothesis is supported by the observation that the lipid-free C-terminal domain of apoAI forms HDL-like structures mediating both cellular cholesterol efflux via the ATP binding cassette transporter A1 (ABCA1) and glucose uptake [35]. In our study, the alterations in core lipids may lead to a conformationally altered apoAI that could negatively influence this apoAI-mediated function of HDLs.

LCAT together with apoAI, the principal catalytic activator, esterifies FC derived from TG-rich lipoproteins or peripheral cells at the surface of discoid HDLs [36]. In our study, HDL-TC gradually increased from normoglycemic patients, to those in prediabetes, and then in T2D; mainly due to the increase in FC, while CE is decreased, resulting in a progressive decrease of the CE/FC ratio. Studies have shown impaired LCAT activity in T2D due to high levels of glycated HDL, which represent poor substrates for this enzyme [37,38], whereas other studies have shown that plasma cholesterol esterification remains unchanged or increased [39,40,41]. Serum TG levels are the major determinant in the regulation of plasma cholesterol esterification [42]. Hypertriglyceridemia, often observed in insulin-resistance states [43], may facilitate LCAT activity [42], therefore possibly confounding the positive association of LCAT activity with T2D. Of note, Riemens et al. showed that this positive association was not observed after adjustment for serum TG levels [27]. The latter can be considered to be in accordance with our findings wherein FC is increased and the CE/TC ratio that reflects the percentage of cholesterol esterification in HDL fraction is significantly diminished. These changes possibly indicate the inefficient LCAT-mediated conversion of FC to CE, with the progressive impairment of glucose homeostasis.

A progressive increase in the TC/PL ratio was observed due to the increase in TC and decrease in PLs. PLs, both glycerophospholipids and sphingolipids, predominate in the HDL lipidome, accounting for 40–60% of total lipid content [44]. These molecules are strong determinants of the structure and architecture of HDLs, as well as their functionality, by modulating surface charge or fluidity or by binding to cellular receptors [44,45]. Thus, alterations in the PL pattern may impair HDL properties. The decrease in HDL-PLs could be the result of the impaired catabolism of atherogenic TG-rich particles due to the decreased transfer of PLs by phospholipid transfer protein (PLTP). Furthermore, hepatic lipase hydrolyzes both TG and PLs, resulting in decreased HDLs’ size and the increased CETP-mediated dissociation of apoAI from HDLs.

The fluidity of the PLs surface monolayer is strongly determined by the amount and proportions of PLs [46] and is particularly important for both scavenger receptor class B member 1 (SR-BI)- and ABCA1-mediated cholesterol efflux [46]. The enrichment of HDLs with PC and SM enhances the bidirectional flux of cholesterol from SR-BI expressing cells to HDL by two different mechanisms: PC increases cholesterol efflux, while SM decreases the influx of cholesterol [47]. However, Schwendeman et al. [48] showed that the ATP binding cassette transporter G1 (ABCG1) and the SR-BI-mediated efflux was higher when HDLs contains SM instead of PC [48]. In the present study, a gradual depletion in both PC and SM was observed from normoglycemic individuals, in patients with prediabetes, and then in T2D patients, possibly indicating a negative implication on the ability of HDLs to efflux cholesterol. Recently, Denimal et al. [49] investigated the alterations in HDL lipid composition of T2D patients compared to controls and the concomitant effect of these changes on cholesterol efflux capacity and the anti-inflammatory properties of these particles. Apart from the common lipid abnormality in HDLs’ core that is the replacement of CE by TG molecules, they found that the FC content was lower in T2D patients compared to controls, whereas the relative proportions of the main HDL choline phospholipids were not statistically different between the two groups. The latter possibly could explain the preservation of the aforementioned properties of HDLs in these patients. PLs are also responsible for the ability of HDLs to inhibit the cytokine-mediated increase in the endothelial cell expression of adhesion molecules [50], reducing the recruitment of blood monocytes into the arterial wall, a process known as anti-inflammatory activity. PC was thought to be the main lipid component responsible for this activity of HDLs [50]. However, HDLs containing SM were more efficient in inhibiting the release of TNF-α, IL-6, and IL-1β compared to those containing PC [48].

In our study, patients with impaired glucose metabolism presented with higher HDL-lysoPC compared to normoglycemic; surprisingly, this increase was more profound in prediabetes compared to T2D. Increased lysoPC potentially reflects the enhanced hydrolysis of HDL-PC by proinflammatory lipoprotein-phospholipases and hepatic lipase. The enrichment of HDL in proinflammatory lysoPC may be relevant to the impaired biological activities of HDL in the patients’ groups [51]. Ståhlman et al. found higher HDL-lysoPC content in dyslipidemic, diabetic patients [52], whereas Cardner et al. found it to be lower [16]. This discrepancy was attributed to the fatty acid bounded to HDL-lysoPC which can be unsaturated fatty acid (anti-inflammatory) [16] or specifically the arachidonic acid (pro-inflammatory) [52].

A gradual enrichment of HDLs in the rest SLs, which represent only a minor fraction of total HDL lipid mass, was observed. De novo ceramide synthesis is increased in insulin-resistant states [53]. Furthermore, ceramide itself, through s positive feedback mechanism, promotes cytokine secretion, resulting in the activation of pro-inflammatory pathways, and also is involved in oxidative stress, inducing mitochondria dysfunction, and promoting cellular apoptosis [53]. HDL-S1P has several beneficial effects on the vasculature; however, glycation can reduce its amount, leading to increased cell death due to the less-effective activation of intracellular survival pathways [54].

A progressive shift of fatty acids from unsaturation toward a saturation state was observed in the present study. The fatty acid pattern of HDL lipids, which reflects both dietary intake and endogenous metabolic processes, closely determines the fluidity of the surface monolayer. PC-fatty acid chain saturation modulates the surface fluidity of HDLs that, as previously mentioned, normally regulates cholesterol efflux capacity [55]. Apart from that, the fatty acid pattern strongly affects endothelial function. SFA induces proinflammatory responses [56], increases endothelial injury, and impairs endothelial repair capacity [57], whereas PUFA improves endothelial function. In addition, the anti-inflammatory activity of HDLs is improved after the consumption of PUFA and reduced when the SFA content of the diet is increased [58]. Linoleic acid, the major fatty acid esterified with FC to form CE gradually decreased, as expected. In addition to linoleic acid, the sum of eicosapentaenoic and arachidonic acid gradually decreased, possibly indicating the low activity of D5-desaturase and the high activity of D6-desaturase [59,60].

Our study has limitations. Data on the height and weight of the participants are not available.

## 4. Materials and Methods

***Subjects:*** Forty patients with prediabetes and 40 patients with T2D (diagnosed on admission) who were attending the outpatient clinics of the University Hospital of Ioannina participated in the study. The control group comprised 40 consecutive individuals who received an annual medical checkup in our clinics and had normal glucose metabolism.

All groups were matched for age, gender, and serum-lipid-profile (total, LDL-, non-HDL- and HDL-cholesterol, triglycerides, apolipoprotein AI (apoAI), and apolipoprotein B (apoB)) to minimize the confounding effect of these parameters on the data analysis (Table 1). Patients with T2D were drug-naive. No individual had evidence of cardiovascular disease according to history, clinical examination, or electrocardiogram. None of the participants were taking lipid-lowering drugs or any other medication known to affect lipid metabolism, including hormonal replacement therapy.

***Diabetes and prediabetes diagnosis***: Τhe diagnosis of T2D was made according to the criteria established by the American Diabetes Association [61]. Patients with prediabetes met the following criteria: (a) impaired fasting plasma glucose, defined as fasting plasma glucose between 100 and 125 mg/dL (5.55–6.99 mmol/L), and/or (b) HbA1c levels between 5.7% and 6.4%. Participants in the normoglycemic group had fasting plasma glucose and HbA1c values below 100 mg/dL and 5.7%, respectively.

***Sample Collection:*** Fasting venous blood samples were obtained in the morning after an overnight fast for all study participants. Serum was separated by centrifugation at 3000× *g* for 15 min for the determination of biochemical parameters, and one 1.5 mL aliquot was stored at −80 °C until NMR analysis.

***Ethics statement:*** The collection of the samples from all participants was conducted in accordance with the guidelines of the Ethics Committee of the University Hospital of Ioannina (Code 78/2018). Written consent was obtained from each participant.

***Determination of Biochemical Parameters***: Serum levels of fasting glucose and lipid parameters were measured on an AU5400 Clinical Chemistry analyzer (Beckman, Hamburg, Germany) by standard procedures. Total cholesterol and triglycerides were determined enzymatically and HDL-cholesterol by a direct assay. LDL-cholesterol was calculated by the Friedewald formula (provided that triglycerides levels were lower than 400 mg/dL or 4.5 mmol/L) and non-HDL-cholesterol by the equation: non-HDL-cholesterol = total cholesterol-HDL-cholesterol. Serum apoAI and apoB were measured by immunonephelometry on a BN ProSpec System (Siemens, Marburg, Germany. HbA1c was measured in ion-exchange HPLC system (Variant II, Bio-Rad Laboratories, Hercules, CA, USA).

### 4.1. HDL Lipidome Analysis by NMR Spectroscopy

***Isolation and lipid extraction of HDL lipoproteins:*** HDL particles were isolated from non-HDL particles by precipitation with dextran sulfate/MgCl_2_, and their lipid content was extracted according to the modified Bligh and Dyer method [62], as described previously [63].

***^1^H NMR spectroscopy:*** The extracted HDL lipids were dissolved in 500 μL of deuterated methanol/chloroform (2:1, *v/v*).All of the NMR spectra were recorded on a 500 MHz Bruker Avance DRX NMR spectrometer (NMR Center, University of Ioannina) operating at a field strength of 11.74 Tesla. A “zgpr” Bruker pulse program was applied with the parameters as follows: 64 scans, 64 K data points with a 5000 Hz spectral width, and a 90° pulse. All free induction decays (FIDs) were multiplied by an exponential weighting function corresponding to the 0.3 Hz line-broadening factor prior to Fourier transformation. NMR spectra were phase- and baseline-corrected and referenced to the methanol signal (δ = 3.30 ppm) using TopSpin 2.1 software (Bruker Biospin Ltd., GmbH, Rheinstetten, Germany).The quantification of HDL lipids was based on the integration of characteristic well resolved signals in the NMR spectrum, corrected for the number of protons and then normalized with respect to the signal from the cholesterol C18 methyl group at 0.68 ppm. The lipid composition of HDLs were expressed as percentages of the total lipid content [63,64].

### 4.2. Statistical Analysis of Data

***Univariate analysis:*** All quantitative data are expressed as mean values ± standard deviation (SDs). Group comparisons were performed using one-way analysis of variance (ANOVA), followed by a least significance differences (LSD) test for pairwise comparisons. A *p*-value < 0.05 was considered to indicate statistical significance.

***Multivariate analysis:*** Principal component analysis (PCA) and orthogonal projections to latent structures discriminant analysis (OPLS-DA) were used to construct pattern-recognition models, to extract the specific atherogenic lipidomic signatures of HDL in groups with different degrees of the impairment of glucose metabolism. The results of the OPLS-DA analysis are displayed on a scores plot (detection observations lying outside the 0.95 Hotteling’s T2 ellipse, grouping trend, or separation). The OPLS-DA model was assessed by goodness-of-fit parameters R^2^ (R^2^X and R^2^Y) and Q^2^, related to the explained and predicted variance, respectively. The cross-validated coefficient of variation analysis of variance (CV-ANOVA) was used to assess the significance of the resulting OPLS-DA model.

## 5. Conclusions

A serum lipid profile provides only a narrow snapshot of the dynamic processes of lipid metabolism. Since metabolic dysfunction is intertwined with the pathophysiology of diabetes and CVD, the detailed characterization of lipid profiling with emerging methodologies such as lipidomic approaches bear a potential to identify patients with subtle (but still proatherogenic) abnormalities that may confer increased risk for progression to overt T2D and/or a higher future CVD risk. It is important to mention that in this regard, the present study not only documents the presence of profound molecular alterations in the HDLs of patients with prediabetes but also equally introduces the notion of glycemic status as a determinant of altered HDL composition. The aforementioned structural modifications and lipid compositional alterations in HDLs may synergistically modify apolipoproteins and enzymes with an adverse impact on the antidiabetic and cardioprotective functions of HDLs.

## Figures and Tables

**Figure 1 metabolites-12-00683-f001:**
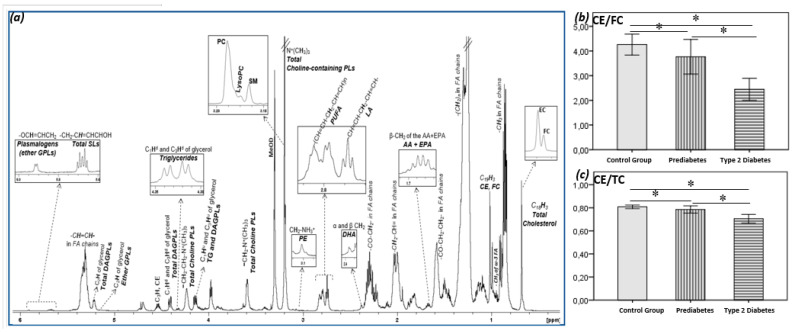
(**a**) ^1^H NMR spectrum of an HDL lipid extract. Figure adopted from [14], (**b**) Cholesterol esters to free cholesterol (CE/FC) ratio in the study groups (mean ± SD), (**c**) cholesterol esters to total cholesterol ratio (CE/TC) in the study groups (mean ± SD). * Statistically significant (*p* < 0.001).

**Figure 2 metabolites-12-00683-f002:**
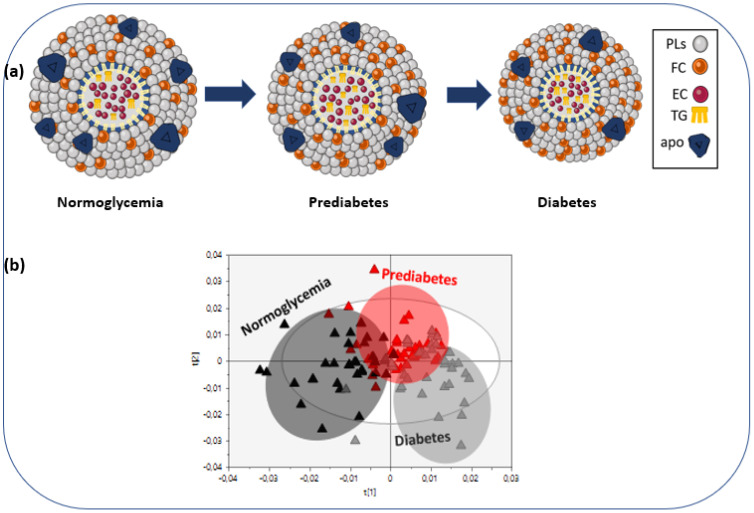
(**a**) HDL particles in the normoglycemic group, patients with prediabetes and type 2 diabetes, (**b**) OPLS-DA scores plot of the HDL lipidomic data from the normoglycemic group (black triangles), patients with prediabetes (red triangles) and those with T2D (gray triangles).

**Table 1 metabolites-12-00683-t001:** Demographic and biochemical characteristics of the study’s participants.

	Normoglycemic Group	Patients with Prediabetes	Patients with T2D
n	40	40	40
Demographic			
Age (years)	55.7 ± 9.6	57.1 ± 7.7	57.2 ± 9.8
Gender (males/females)	21/19	20/20	22/18
Biochemical			
Total cholesterol (mg/dL)	184 ± 26	178 ± 23	184 ± 40
Triglycerides (mg/dL)	100 ± 34	113 ± 43	109 ± 37
HDL-cholesterol (mg/dL)	49 ± 11	51 ± 9	47 ± 8
LDL-cholesterol (mg/dL)	115 ± 21	104 ± 24	116 ± 35
non-HDL-cholesterol(mg/dL)	135 ± 24	127 ± 24	137 ± 37
apoAI (mg/dL)	139 ± 23	160 ± 21	136 ± 23
apoB (mg/dL)	79 ± 18	78 ± 20	87 ± 20
Glucose (mg/dL)	90 ± 7	106 ± 8 *	157 ± 24 *#
HbA1c (%)	4.9 ± 0.7	5.9 ± 0.4 *	7.3 ± 0.7 *#

* *p* < 0.001 compared to the normoglycemic group # *p* < 0.001 compared to patients with prediabetes. Color was used to distinguish the three groups from each other.

**Table 2 metabolites-12-00683-t002:** HDL composition of major lipid classes.

	Normoglycemic Group	Patients with Prediabetes	Patients with T2D
Mean ± SD	Mean ± SD	vs. Normoglycemic	Mean ± SD	vs. Normoglycemic	vs. Prediabetes
% Change	*p* Value	% Change	*p* Value	% Change	*p* Value
**Cholesterol, Total**	**40.22 ± 1.98**	**41.38 ± 2.49**	**+2.88**	**<0.05**	**41.74 ± 2.42**	**+3.78**	**<0.01**	**+0.87**	**NS**
Free	7.70 ± 0.79	8.93 ± 1.75	+15.97	<0.01	12.33 ± 1.93	+60.13	<0.001	+38.07	<0.001
Esterified	32.52 ± 1.57	32.45 ± 1.21	−0.22	NS	29.41 ± 1.96	−9.56	<0.001	−9.37	<0.001
**Triglycerides (TG)**	**4.28 ± 0.71**	**5.42 ± 0.85**	**+27.17**	**<0.001**	**5.83 ± 1.37**	**+36.53**	**<0.001**	**+7.37**	**NS**
**Phospholipids (PLs), total**	**55.50 ± 2.10**	**53.20 ± 2.16**	**−4.14**	**<0.001**	**52.43 ± 3.15**	**−5.53**	**<0.001**	**−1.45**	**NS**
Core lipids, total	36.79 ± 1.77	37.88 ± 0.97	+2.96	<0.05	35.24 ± 2.71	−4.21	<0.01	−6.97	<0.001
Surface lipids, total	63.21 ± 1.77	62.12 ± 0.97	−1.72	<0.05	64.76 ± 2.71	+2.45	<0.001	+4.25	<0.001
TC/PLs	0.73 ± 0.06	0.78 ± 0.08	+6.85	<0.01	0.80 ± 0.11	+9.59	<0.001	+2.56	NS
CE/TG	7.81 ± 1.38	6.12 ± 0.94	−21.64	<0.001	5.34 ± 1.43	−31.63	<0.001	−12.75	<0.01
	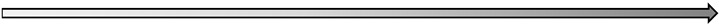

Values are expressed in percentages of total lipids (mol/100 mol of total lipid content) and are means ± SD. Bold: Different lipid class. Color in graduated shade was used to be in accordance with the pro-gressive changes occurred in HDL lipid composition in normoglycemic individuals, to prediabetes and then to diabetes.

**Table 3 metabolites-12-00683-t003:** Phospholipid profiling of HDL lipoproteins.

	Normoglycemic Group	Patients with Prediabetes	Patients with T2D
Mean ± SD	Mean ± SD	vs. Normoglycemic	Mean ± SD	vs. Normoglycemic	vs. Prediabetes
% Change	*p* Value	% Change	*p* Value	% Change	*p* Value
**Glycerophospholipids (GPLs), total**	43.67 ± 2.55	41.07 ± 2.38	−5.95	<0.001	40.71 ± 2.93	−6.78	<0.001	−0.88	NS
Phosphatidylcholine (PC)	32.70 ± 2.27	32.18 ± 2.98	−1.59	NS	31.03 ± 2.80	−5.11	<0.01	−3.57	NS
Lysophosphatidylcholine (LysoPC)	2.69 ± 0.54	4.36 ± 0.64	+62.08	<0.001	4.06 ± 0.91	+50.93	<0.001	−6.88	NS
Phosphatidylethanolamine (PE)	1.05 ± 0.25	0.70 ± 0.28	−33.33	<0.001	0.73 ± 0.20	−30.48	<0.001	−4.28	NS
Phosphatidylinositol (PI)	1.78 ± 0.53	1.11 ± 0.33	−37.64	<0.001	2.34 ± 0.74	+31.46	<0.001	+110.81	<0.001
Rest GPLs ^a^	5.45 ± 1.55	2.72 ± 1.21	−51.18	<0.001	2.55 ± 0.77	−49.16	<0.001	+4.14	NS
**Ether glycerolipids (ether GLs), total**	**5.40 ± 0.88**	**4.92 ± 0.63**	**−8.89**	**<0.05**	**4.29 ± 0.95**	**−20.56**	**<0.001**	**−12.80**	**<0.01**
Plasmalogens	1.60 ± 0.32	1.72 ± 0.59	+7.50	NS	1.52 ± 0.25	−5.00	NS	−11.63	<0.05
Rest ether GLs ^b^	3.80 ± 0.85	3.20 ± 0.74	−15.79	<0.01	2.77 ± 0.90	−27.11	<0.001	−13.44	<0.05
**Sphingolipids (SLs), total**	**6.43 ± 0.95**	**7.21 ± 1.02**	**+12.13**	**<0.001**	**7.43 ± 0.93**	**+15.55**	**<0.001**	**+3.05**	**NS**
Sphingomyelin (SM)	6.11 ± 0.90	5.51 ± 1.08	−9.82	<0.01	5.48 ± 0.80	−10.31	<0.01	−0.54	NS
Rest SLs ^c^	0.32 ± 0.11	1.70 ± 0.77	+413.25	<0.001	1.95 ± 0.90	+509.38	<0.001	+14.71	NS
PC/SM	4.75 ± 0.54	5.37 ± 1.23	+13.05	<0.01	4.60 ± 0.89	−3.16	NS	−14.34	<0.001
	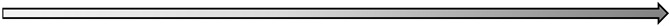

Values are expressed in percentages of total lipids (mol/100 mol of total lipid content) and are means ± SD. a: mainly phosphatidylserine, phosphatidylglycerol, b: mainly PAF, c: mainly ceramide. Bold: Dif-ferent phospholipid class. Color in graduated shade was used to be in accordance with the pro-gressive changes occurred in HDL phospholipid profiling in normoglycemic individuals, to pre-diabetes and then to diabetes.

**Table 4 metabolites-12-00683-t004:** Fatty acid profile of HDL lipoproteins.

	Normoglycemic Group	Patients with Prediabetes	Patients with T2D
Mean ± SD	Mean ± SD	vs. Normoglycemic	Mean ± SD	vs. Normoglycemic	vs Prediabetes
% Change	*p* Value	% Change	*p* Value	% Change	*p* Value
**% Saturated**	**37.07 ± 3.11**	**41.87 ± 1.81**	+12.95	<0.001	**47.23 ± 6.78**	+27.41	<0.001	+12.80	<0.001
**% Unsaturated**	**62.93 ± 3.11**	**58.13 ± 1.81**	−7.63	<0.001	**52.77 ± 6.77**	−16.14	<0.001	−9.22	<0.001
**% Monounsaturated**	**10.67 ± 3.68**	**6.35 ± 2.45**	−40.49	<0.001	**5.40 ± 2.30**	−49.39	<0.001	−14.96	NS
**% Polyunsaturated**	**52.26 ± 2.78**	**51.78 ± 3.36**	−0.92	NS	**47.37 ± 5.73**	−9.36	<0.001	−8.52	<0.001
▪ **Linoleic acid**	**19.39 ± 2.24**	**18.73 ± 1.49**	−3.40	NS	**16.65 ± 2.04**	−14.13	<0.001	−11.11	<0.001
▪ **Eicosapentaenoic + arachidonic acid**	**12.26 ± 1.68**	**10.71 ± 1.63**	−12.64	<0.001	**10.14 ± 1.75**	−17.29	<0.001	−5.32	NS
▪ **Docosahexaenoic acid**	**4.17 ± 0.68**	**2.47 ± 0.41**	−40.77	<0.001	**3.29 ± 0.82**	−21.10	<0.001	+33.20	<0.001
**Saturated/unsaturated**	**0.59 ± 0.09**	**0.72 ± 0.05**	+22.03	<0.01	**0.93 ± 0.26**	+57.63	<0.001	+29.17	<0.001
**Saturated/polyunsaturated**	**0.71 ± 0.08**	**0.81 ± 0.09**	+14.08	<0.05	**1.03 ± 0.28**	+45.07	<0.001	+27.16	<0.001
	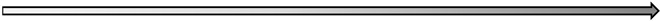

Values are expressed in percentages of total lipids (mol/100 mol of total fatty acids) and are means ± SD. Bold: Different FA class. Color in graduated shade was used to be in accordance with the progres-sive changes occurred in HDL fatty acid composition in normoglycemic, to prediabetes and then to diabetes.

## Data Availability

Data are available on Kostara, C.E.; et al. Progressive, qualitative and quantitative alterations in HDL lipidome from healthy subjects to patients with Prediabetes and Type 2 Diabetes. *Dryad. Data Set ProMED-Mail Website*
**2022**. https://doi.org/10.5061/dryad.t76hdr83q.

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
