# Peer review of "Progressive, Qualitative, and Quantitative Alterations in HDL Lipidome from Healthy Subjects to Patients with Prediabetes and Type 2 Diabetes"

_metabolites, 2022, doi:10.3390/metabo12080683_

Round 1

Reviewer 1 Report

This is an interesting and valuable study for the field. I have several comments to improve its quality, hopefully.

Please provide the statistical tests for all the clinical covariates listed in Table 1. It is crucial to examine if the matched design (age and gender) works well. Besides, since height and weight are important confounding factors, they should be evaluated if data is available. Otherwise, it should be stated as a limitation of this study.

Interestingly, OPLS-DA was used and visualized for the 3-group design. It is commonly used for 2-group settings because a 3-group design makes it hard to correctly interpret, especially for the % of accurate class prediction. Besides, not sure how was the label used for the calculation maximizes the co-variance among groups, I.e., class order was taken into account in the PLS regression (label-sensitive vs. label-non sensitive). These must be provided as they may change the interpretation significantly. I recommend the team try analyzing the data in the three pair-wise OPLS-DA models and report the results as supplementary data.

In the ethics statement, please add the number/code of the IRB approval.

I would encourage the authors to deposit at least the processed data to a public repository to allow the re-use of the data.

The submitted file does not comply well with the template format used by the journal. Besides, minor typos/formatting errors should be fixed (e.g., MgCl2)

Author Response

We thank the reviewer for the kind consideration of our work and his/her suggestions. Please find our responses in the attached file.

Reviewer 2 Report

This study investigates the HDL lipidome to unravel the metabolic pathways underlying diabetes progression.

The study is well conducted and, although no significant changes are observed, some results are different between the groups studied. It may be an advance in the search for biomarkers that allow the detection of people with a prediabetes state.

It is possible, if the number of samples was increased, to obtain more representative results that show differences between a state of prediabetes and diabetes.

It is a good study, although in the future it would be necessary to extend the results described to more characteristic markers of the development of diabetes.

Author Response

We thank the reviewer for the kind comments of our work.

Round 2

Reviewer 1 Report

The quality of the manuscript has improved. I expected more regarding the information to be described in the limitations that may sharpen the subsequent investigations.